# Phase-Changing Glauber Salt Solution for Medical Applications in the 28–32 °C Interval

**DOI:** 10.3390/ma14237106

**Published:** 2021-11-23

**Authors:** Linus Olson, Carina Lothian, Ulrika Ådén, Hugo Lagercrantz, Nicola J. Robertson, Fredrik Setterwall

**Affiliations:** 1Department of Women’s and Children’s Health, Karolinska Institutet, 17177 Stockholm, Sweden; ulrika.aden@ki.se (U.Å.); hugo.lagercrantz@ki.se (H.L.); 2Department of Neonatology, Vietnam National Children’s Hospital, Hanoi, Vietnam; 3Neonatal Unit, Stockholm Söder Hospital, 11883 Stockholm, Sweden; carina.lothian@regionstockholm.se; 4Institute for Women’s Health, University College London, London WC1E 6HU, UK; n.robertson@ucl.ac.uk; 5Division of Energy Processes, Chemical Engineering and Technology, Royal Institute of Technology, 10044 Stockholm, Sweden; Fredrik.setterwall@comhem.se

**Keywords:** glauber salt, phase change material, neonatal asphyxia, hypoxic ischemic encephalopathy, therapeutic hypothermia

## Abstract

(1) Background: The field of medicine requires simple cooling materials. However, there is little knowledge documented about phase change materials (PCM) covering the range of 28 to 40 degrees Celsius, as needed for medical use. Induced mild hypothermia, started within 6 h after birth, is an emerging therapy for reducing death and severe disabilities in asphyxiated infants. Currently, this hypothermia is accomplished with equipment that needs a power source and a liquid supply. Neonatal cooling is more frequent in low-resource settings, where ~1 million deaths are caused by birth-asphyxia. (2) Methods: A simple and safe cooling method suitable for medical application is needed for the 28 to 37.5 °C window. (3) Results: Using empirical experiments in which the ingredients in Glauber salt were changed, we studied the effects of temperature on material in the indicated temperature range. The examination, in a controlled manner, of different mixtures of NaCl, Na_2_SO_4_ and water resulted in a better understanding of how the different mixtures act and how to compose salt solutions that can satisfy clinical cooling specifications. (4) Conclusions: Our Glauber salt solution is a clinically suited PCM in the temperature interval needed for the cooling of infants suffering from asphyxia.

## 1. Introduction

Phase Change Materials (PCMs) are used for energy storage [1,2,3,4], heating and cooling of buildings [5,6], optimization of different residential climates [7,8,9,10], as well as cooling of computers and other telecom equipment [11]. PCM can also be used for medical purposes [12,13], which is the focus of the present investigation. For reviews of the field and relevant inventions, see the IEA Annex 17 Final report [14] and Zalba et al. [15].

### 1.1. Clinical Background 

Newborn infants may suffer brain damage from asphyxia and oxygen deprivation during delivery. The medical term often used for this condition is Newborn Hypoxic Ischemic Encephalopathy, nHIE. By controlled therapeutic hypothermia (cooling of the human body) to a stable temperature of 33.5 ± 0.5 °C, children at risk of nHIE have been shown to have a reduced risk of death or severe disability at 18–22 months follow-up, based on several randomized, controlled investigations [16,17,18,19], after extensive pre-studies [20,21,22]. Nevertheless, there is a need to further investigate and assess several aspects of this complex therapy [19,23,24,25], and not to cool too much [26,27], as well as to clarify how the brain responds and how the effects of cooling in combination treatments, such as addition of Xenon gas, operate [28]. A key factor influencing the therapeutic effect of hypothermia is the interval between the insult and the induction of hypothermia [20,21]. To obtain maximum benefit, postnatal hypothermia should be started as soon as possible [29,30], and within 6 h after the insult [20,29]. When treatment is started as late as after 12 h, only limited effects should be expected. A meta-analysis stresses the need for further studies, and specifically points to the need to study earlier initiation of hypothermia [23,24]. Conventional cooling to prevent nHIE is based on circulating cool water and requires electricity. To allow fast initiation of cooling, e.g., during transport of patients, as well as to provide a system that is independent of both water and electricity, PCMs have been considered. PCMs melt within given temperature ranges, and are used, for example, for thermal energy storage [1,2,31,32].

### 1.2. PCM Background

Thermal energy can be stored in multiple ways. In PCM, the latent heat of the material is used [5]. Each time a material changes phase between solid, liquid and gas, heat is disseminated or taken up. Ice constitutes a simple form of PCM, used since the beginning of modern history [33]. Its PCM characteristics can be exemplified by the fact that the amount of energy needed to melt 1 kg of ice is 80 times larger than the energy needed to increase the temperature of the same amount of ice by 1 degree [31]. However, ice undergoes its phase change at a temperature that is unsuitable for the cooling of infants. Other more commonly used PCMs are often based on waxes (fatty acid and ester or paraffins, such as octadecane), eutectic salt mixtures, and salt hydrates. Figure 1. PCMs for our purpose should have phase change temperatures that allow them to be used to cool babies via direct skin contact, and should also be rechargeable (i.e., re-frozen) at room temperature. Advantages of PCMs include the ability to maintain a constant temperature during the phase change, thus minimizing fluctuations and providing better stability in the energy exchange process. PCMs also have a high energy storage density, making them ideal for objects that need to be constantly cooled or heated for a prolonged period by energy removal or delivery, respectively.

Progress in absorption systems (a subset of systems for heating and cooling of buildings) has included peak temperature shaving of systems that generate too much heat or cold during limited time periods. In recent decades, there have been evolutions in the research field, including passive cooling of buildings [13,15], energy saving methods, absorption systems (a subset of heating and cooling buildings). One way to solve this problem is to microencapsulate PCMs, allowing the mixing of PCMs with other materials or the flux of PCMs [34,35,36]. In addition to common buildings, potential uses for PCMs include heating and cooling of greenhouses to adjust climate and improve production, the multiple needs for heating and cooling in the medical world, transportation and conservation storage containers, and in computers and other telecom equipment [10].

The energy density in sensible heat storage materials is given by the temperature range and the heat capacity of the storage material [37]. The temperature range depends on the application at hand, and is limited by the temperature of the storage material as well as the temperature of the heat source, in our case the infant. The energy density, in our case the capacity to remove heat from a newborn baby, is highest when using a PCM with a phase change temperature similar to the desired cooling temperature [15,38,39,40].

Within a given temperature interval ∇T = T_2_−T_1_, the sensible and latent stored heat Q_lat_ can be calculated as:Q_lat_ = T_1_∫T_2_ cp × dT + H_ls_
where H_ls_ is the heat of the fusion at the phase change temperature T_ls_.

The attainable temperature difference ∇T is dependent on the charging temperature Tc, given by the heat source. Depending upon whether the temperature of the energy source is higher or lower than the ambient temperature, the storage material will work as a heat or a cold sink. This means that insulation of the storage is needed to protect the storage material from losses over time. Preliminary results of studies were presented at the Annex 17 meetings, and a summary is found in the IEA final report, as well as in [14]. The general idea behind the current paper was first presented at the IEATA conference in 2004 in Arvika, Sweden [38]. Preliminary studies were performed in China and Turkey [41], and preliminary medical conclusions were drawn in Stockholm, Sweden on the basis of animal experiments [39].

### 1.3. Experimental Background

The PCMs intended for our purposes are solid at room temperature; when in contact with a warmer object, these PCMs absorb and store heat, ultimately liquefying. Conversely, liquid PCMs can solidify when giving off heat, thus working as heat buffers and stabilizing the temperature of objects in contact with them. The objective of this study was to evaluate different mixtures of Glauber salt (named after Johann Rudolf Glauber, who discovered it in 1625), in order to develop a mixture with a temperature range of 28–32 °C that is suitable for the medical field. The material should also be such that no outside force would be needed to start or stop the energy transfer process between the PCM and the object to be cooled or heated. It should also be independent of electric power and water in terms of reversing the phase transition.

## 2. Materials and Methods

For medical uses, the studied PCMs should meet a list of 10 important criteria, described in Table 1.

These criteria eliminate all highly toxic materials, as well as compounds with very slow energy transfer characteristics or very low energy storage capacity. Based on these criteria, and discussions with expertise in material physics from the Royal Institute of Technology in Stockholm, we selected Glauber salt in different forms for further study. In reaching this decision, we took into account the mixture-processing expertise available to our project through PCM producers TST AB Sweden and its subsidiary, Climator AB (a PCM producer in Sweden) and Rubitherm. A shared knowledge protocol was established, which included the intention to produce test amounts of suitable PCMs for medical use.

We used a T-History-like method [42,43] to evaluate the findings. To generate the different materials, we used the well-known Glauber salt Na_2_SO_4_ 10H_2_O, which has a phase change temperature point of 32 °C, and mixed it with NaCl, in an attempt to change the temperature stabilization point and produce PCM that would have a lower set point, e.g., at 30 °C. For equipment see Appendix A.

The mixtures were constructed as follows: a well-defined amount of Na_2_SO_4_, 10.33 g, was first measured. Next, NaCl was added. The salt combination was then mixed with water to obtain the desired NaCl and Na_2_SO_4_ mixtures. The amount of NaCl was varied so that, on a dry weight basis, the percentage of NaCl was between 1.5 and 7% in 0.5% increments. The solution was stirred while being heated until all salts were completely dissolved. To stabilize the solution and make it more convenient to handle in future commercial applications, 1–5 g of cethyl methyl cellulose (CMC) was added as a gelifier agent. The CMC also allowed different salt hydride products to bind different amounts of water, during different stages of the phase change to avoid separation due to gravity. Likewise, the presence of CMC was also expected to keep the NaCl better distributed within the solution. The complete solution was then poured into glass test tubes for experimental usage. The temperatures of the glass tubes were monitored (DacPad, National Instruments, Austin, TX, USA) using temperature probes placed in the middle of the test solutions in the glass tubes and connected to a data logger. The glass test tubes were repeatedly cooled and heated using a water bath (Lauda RE106 with heater, The Lab World Group, Hudson, MA, USA) /cooler Lauda E100, (The Lab World Group, Hudson, MA, USA). The temperature signals were stored and monitored using a computer with appropriate software (Medical Cooling, Labview, version 8.2) (configured using Labview). The system allowed temperature control, control of sampling rates (time between measurements), and the use of up to four input channels. A monitor was used to display temperature curves in real time, to help evaluate the results and to aid storage of results. The four different channels were used to measure glass test tubes with contents as follows: channel 1: jellified water (control); channel 2: water; channel 3: experimental PCM mixture of interest; channel 4: known PCM mixture for calibration of the experiment. The experimental setup with test tubes in a water bath is schematically shown in Figure 2.

For each mixture we heated the glass test tubes (10 mL) in a water bath to 45 °C and then lowered them to 5 °C and then cycled this procedure at least 3 times. This strategy was used to ensure the elimination of any problems of non-homogeneous mixtures and the risks of crystallization irregularities of different mixtures such as the presence of Na_2_SO_4_ 7H_2_O instead of Na_2_SO_4_ 10H_2_O, and other faults that might compromise reproducibility. Because of the relatively fast temperature changes of the water bath, we added semitransparent insulation sheets around the test tubes to slow down the temperature change process of the PCM somewhat. This also helped prevent glass tubes closer to the heat source from heating up faster than tubes further away.

## 3. Results

Adding different amounts of NaCl was found to modify the melting temperature of the test solutions (Figure 3). From a melting point of 32 °C in the absence of NaCl, additions of up to 5% NaCl decreased the melting point of the mixture to about 25 °C, while further additions of NaCl had little effect on the melting point. We visualized the samples melting points, dissolution/crystallization point as well as usage of the T-history over time in figure after the first two runs (to avoid initial mixing interfering with results).

The salt solutions have temperature change curves that differ from the water curve when warmed up to about 28 °C, and that flattens out briefly around 30–32 °C, depending on the mixture. This flattening occurs when the salt takes up energy from the surroundings before being heated again as expected. This is illustrated in the first part of the curves in Figure 4. When being cooled, the different materials were less diverse, but even here a small variation in the slope of the curves could be noted, as seen in the right part of Figure 4. In mixtures with a small percentage of NaCl, there was a risk for crystallization with 7 instead of the intended 10 H_2_O molecules. When temperatures were lowered to 5°, this caused a temporary rise of temperature to 10 or 12 °C instead of a smooth curve (here seen as a peek in Figure 4 in the very end). The salt then acted in a non-regulated way until the solution was heated up again. All curves can be plotted into a phase–temperature diagram for water NaCl Na_2_SO_4_ mixture. Such a diagram is shown in Figure 5.

Our results in the area of 0–10% NaCl in a solution of 30% Na_2_SO_4_ (5g Na_2_SO_4_, H_2_O, 0.833 g CMC) confirm previously published phase diagrams for temperature vs. salt composition (Figure 5b (25 °C)). We allowed a 0.2–0.3-degree tolerance between measurements, because the reliability of the water bath servo control was ±0.2 °C, and in order to cope with other forms of variability, such as forms of crystallization. If a particular salt solution is dissolved with a 7- instead of 10-crystal water configuration, the next cycle will be affected. Additionally, our sample rate was set to every 10 s (to keep data files manageable). Therefore, curves shown in the figures may not reveal small variations occurring between temperature measurements and cycles. Comparing our data to Figure 6, our curve does not appear to fit the published diagram so well. This is due to our curve seeming to plateau, but as indicated in the curve drawn on top, the curve actually continues descending when higher amounts of NaCl is mixed into the solution. This is known, because other mixtures on the market for Glauber salt contains more NaCl, e.g., ClimSelC 21 from Climator. The faulty point at 2% might be due to a faulty probe, but since the curve is not a straight line superimposing the different diagrams from Figure 6, as well as other similar curves, suggests our double bent one might have hit a low point in our measurements. By altering the amount of CMC within limits of 3 to 8%, it was possible to alter the viscosity of the gel. However, as long as the difference in concentration between the two salts did not change, the behavior of the curve did not change much. Thus, depending on the specifications for a certain application, one can alter viscosity to adapt to specific needs. However, if the thickness of the gel is altered too much, PCM compositions with low amounts of NaCl might become inhomogeneous, leading to different behavior in different compartments.

## 4. Discussion

The investigation of different mixtures of NaCl, Na_2_SO_4_ and water in a controlled way resulted in a better understanding of how these different mixtures act in terms of phase change characteristics when subjected to rising and falling temperatures. This has made it possible for us to identify a particular composition of salts that can be reproduced to fulfill the specifications of medical needs.

The risk of small particles interfering with the procedure and material properties is higher when the mixture contains smaller amounts of NaCl in the solution. The risk of the anhydrite solution was avoided in our experiments by positioning our mixture of interest to the left of the normal Glauber salt mixture seen in the concentration–weight fraction of the Na_2_SO_4_ diagram (Figure 6).

The anhydrite could have caused problems, since variation of NaCl content would then be of less importance in the behavior of the mixture in our temperature interval. Using 30% Na_2_SO_4_ as the base for our mixture, we were able to avoid sodium sulfate deposition, which could otherwise cause additional problems at all stages. Any deposition precipitate in the mixture would act as a foreign particle, which would increase the risk of crystallization at temperatures other than the desired ones, as well as altered amounts of different chemicals in the mixture, available to react as planned in the remaining solution. An uncontrolled process could lead to more or less cooling, and to more or less deposition, further adding to an unstable situation. Some of this has been described, and solutions have been suggested, by Mehling and Cabeza [44]. One way to solve this problem is to microencapsulate PCM, allowing the mixing of PCM with other materials or flux of PCM. However, we mixed Na_2_SO_4_, NaCl and CMC to find new materials in our desired temperature range; findings that have not previously been reported.

Clinically, controlled cooling with circulating water has become a standard treatment of newborn infants at risk of brain damage due to oxygen deprivation. Accumulating clinical experience with excessively cooled children or infants suggests that prolonged periods with unstable temperatures may aggravate brain damage. [18,19,26]. In infants in need of cooling, rectal temperatures may vary from 31 to 38 °C during the initial day of treatment, due to intrinsic temperature regulation mechanisms and environmental factors. These factors may continue to cause fluctuations in temperature that might hamper the final effect of the hypothermia, and therefore, a stable temperature should be maintained throughout the treatment period. On the basis of animal work, it is also known that rewarming too rapidly might have negative effects on neurological outcome. It is, therefore, a significant advantage that PCM mattresses avoid these extreme variations, and that the target temperature remains stable during the entire cooling period. In animal experiments, we have demonstrated that PCM acts like an energy buffer, leading to fewer temperature fluctuations, and hence a more stable temperature, since our salt hybrid gives energy back to the animal when the temperature is too low and absorbs energy when the temperature is above the phase change temperature [45]. Our experimental use of PCM mattresses (Figure 7 and Figure 8) is supported by another advantage of using PCMs for cooling: the inability of a temperature undershoot occurring during the intended cooling period, eliminating such problems originating from human error when manually regulating the temperature of the cooling.

Medical use of the knowledge obtained in this study could help avoid unstable temperatures, which cause harm (swing of temperatures has been known to counteract the hypothermia) [46], and can cool down or obtain temperatures in a simple way, also making it useful for the transport of patients to hospitals, where treatment can be continued. It can also help when starting treatment of newborns at risk of hypoxic ischemic encephalopathy or asphyxia within the 6 h treatment start window in order to prevent brain or organ injuries in remote places [47]. A mattress with PCM does not require an electricity source or water during transport to a high-level centers where newborns can be treated in the same way as the standard state of treatment in western societies. 

The material knowledge can also be used to keep patients of different ages within a normal temperature interval in areas with high outside temperatures, highly variable temperatures, or during transport, as well as together with textiles with encapsulated PCM [48,49,50]. These textiles can then be used in blankets or for the staff to maintain normothermia during transport in warm countries.

Further studies into the use of PCM may lead to applications in different medical disciplines, as well as for the transport of samples or isolates (human, animal and environment samples/isolates), including bacteria and virus samples, blood samples, vaccines, etc. [51]. A Glauber solution seems to be a promising way forward for medical use.

## 5. Conclusions

A PCM solution, in our case a Glauber salt solution, can function as a medically well-suited phase change material in the temperature interval needed for cooling of infants suffering from oxygen deprivation during birth or within 6 h. Such a Glauber salt mixture should keep the set rectal temperature of the infant at 33–34 ± 0.5 °C. We confirmed and tested different amounts of NaCl added to the Glauber salt mixture and found that by altering the percentage we could construct PCMs for all the specified temperatures in the interval of 20–32 °C. By using only 30% instead of the normal amount of Glauber salt (Na_2_SO_4_ × 10 H_2_O) as our starting material, we could avoid problems caused by sulfate anhydrites as well as precipitates in the solution. In this way, we also obtained a more liquid mixture in which all components involved could be better mixed before adding CMC. We were thus able to study the involvement of different amounts of NaCl better than if we would have chosen a higher relative amount of Na_2_SO_4_. Based on the present results, we are confident that optimal mixtures can be developed for cooling and heating purposes, not the least in the medical field.

From a medical point of view, we were interested in the development of technically simple solutions to the problem of brain cooling, applicable in remote sites and countries lacking advanced medical resources [46,52,53]. Such a solution should allow treatment soon after birth and improve the possibility of already starting safe cooling during transportation. A single mattress can be used repeatedly at least 50 times. Clinical use of the knowledge in this article led us to produce a mattress in Sweden and use a PCM mattress in Vietnam and Sweden. This also tests whether the knowledge obtained about PCM worked as expected in a climate where higher temperatures are occurring than in Europe, and larger temperature swings exist. The mattress was used and seemed to work as well as or better than other methods for hypothermia treatment of children with HIE [47]. Additionally, globally, e.g., in India, PCM mattresses have been successfully used even though there were problems when temperatures went over 33 °C [52,54,55,56]. The general conclusion is that PCMs are very useful for medical applications both in high- and in low–middle-income settings, in hospitals and during transportation, and can be used if the material is constructed correctly for the intended temperature or temperature interval in many situations where steady controlled cooling is needed.

## Figures and Tables

**Figure 1 materials-14-07106-f001:**
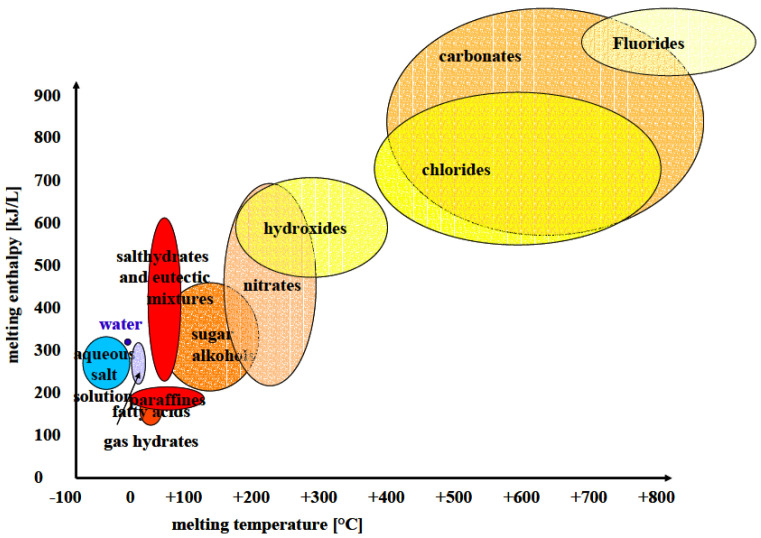
Available phase changing materials and the relation between melting temperature and heat of fusion from these materials.

**Figure 2 materials-14-07106-f002:**
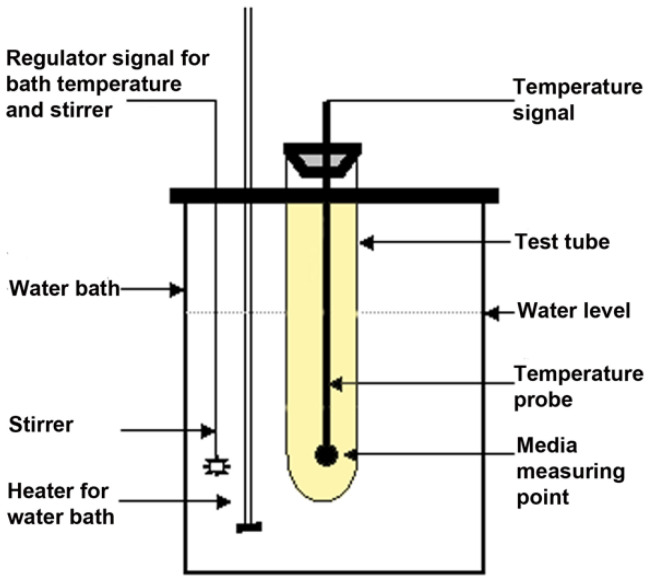
Schematic illustration of experimental setup for monitoring temperature characteristics of different PCM mixtures (yellow) in a water bath with temperatures cycling between 5 and 45 °C. A temperature probe is located in the material to be tested. The test tube has a thin outside insulation (not shown) to dampen speed of temperature changes (see text). The water bath contains a heater, and a stirrer. The stirrer also carries a temperature probe, such that both the heater and the stirrer rpm is controlled by water temperature and temperature changes. Control test tubes (not shown) contain water, jellified water or a known PCM mixture.

**Figure 3 materials-14-07106-f003:**
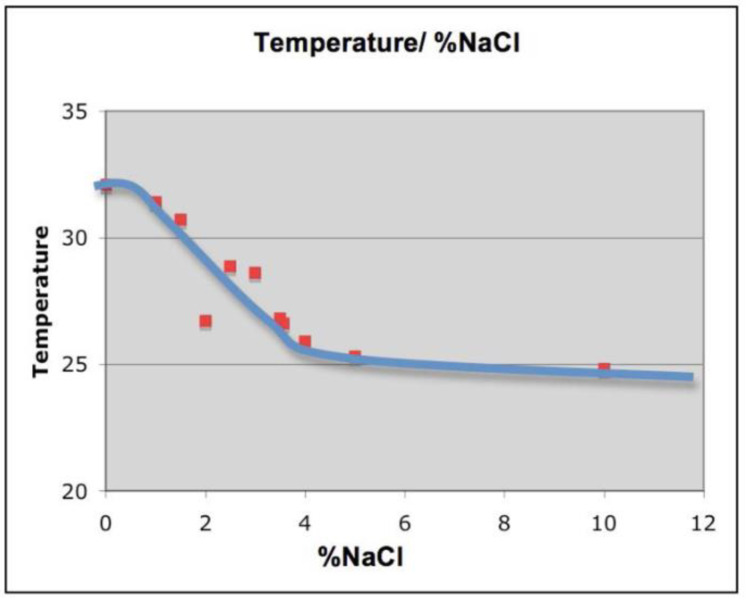
Effect of adding different concentrations of NaCl on melting temperature of PCM. In the absence of NaCl, the melting point (red) is 32 °C. Adding increasing amounts of NaCl decreases the melting temperature to approximately 25 °C when 5% NaCl is added. Further additions of NaCl do not appear to cause further decrease of the melting point. The approximated curve (blue) does not illustrate the possibility that there may be further complexity of the melting point temperatures in the 1.5–3.0% NaCl range, due to the technical limitations of our experimental set up. However, the correlation is significant.

**Figure 4 materials-14-07106-f004:**
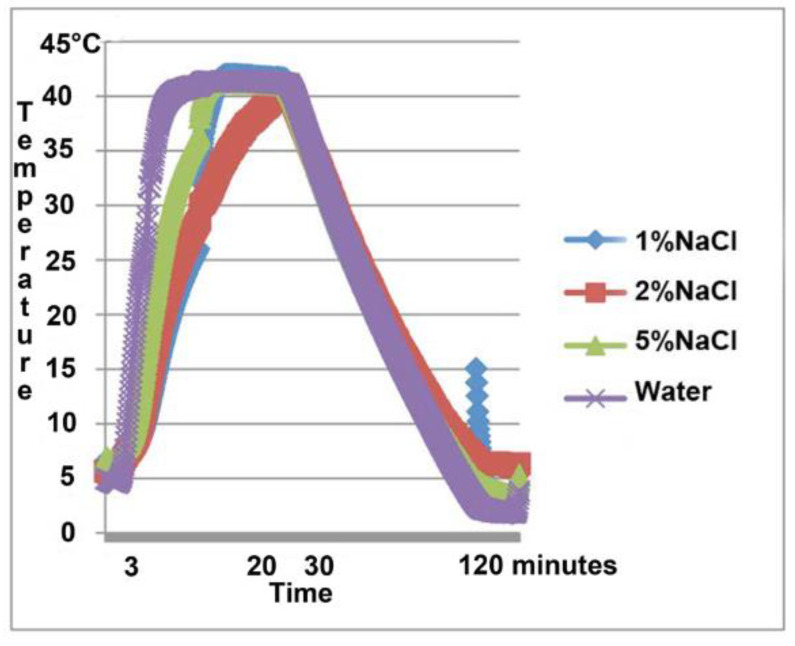
Heating–cooling curves for different mixtures of NaCl and Na_2_SO_4_ in water with CMC added as stabilizing gelifier. The PCM mixtures are compared to water, also gelified by CMC (purple). During heating, all three PCM mixtures differ from water by having a lesser slope and also by brief periods of energy take up at around 30–32 °C (1% NaCl, blue curve is partly hidden). During cooling, there is less diversity, although the slopes of the PCM curves are less steep than that of water. Please note that lowering the bath temperature to 5 °C caused a temporary increase of temperature in PCM containing 1% NaCl (blue).

**Figure 5 materials-14-07106-f005:**
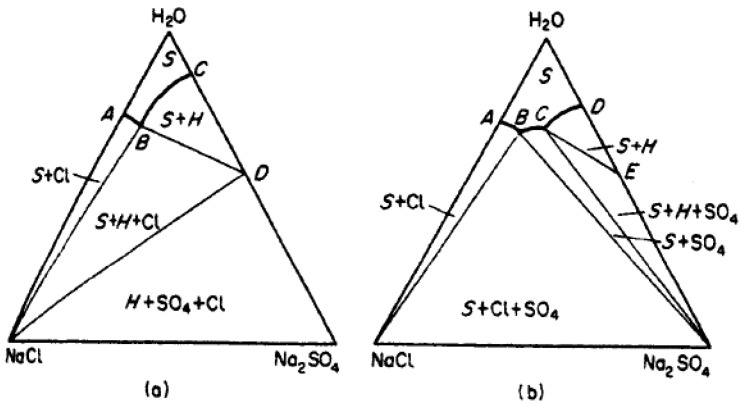
Phase diagrams for the system NACl-Na_2_SO_4_- H_2_O at 17.5 (**a**) and 25 °C (**b**). From Mullin, J.W.: Crystallization, Fourth Edition, Reed Educational and Professional Ltd. Reproduced with permission from Butterworth-Heinemann, Oxford. Please note that the B-C curve descends with increasing amounts of NaCl. Na = Natrium, S = sulfite, Cl = Clorine, H = helium, O = Oxygen.

**Figure 6 materials-14-07106-f006:**
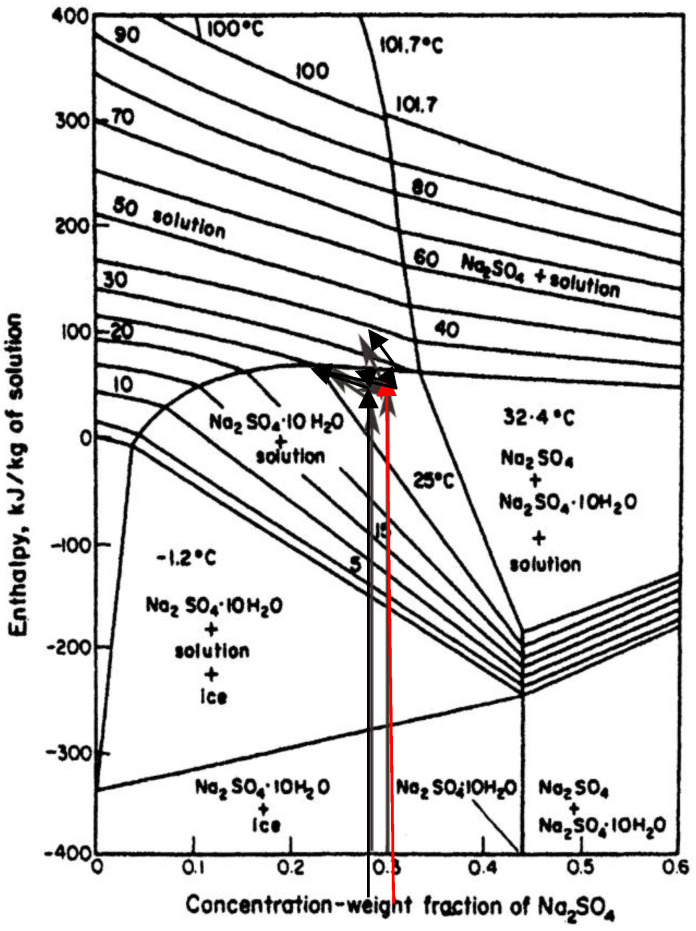
Enthalpy–concentration weight fraction temperature diagram from Mullin, J.W.: Crystallization, Fourth Edition, Reed Educational and Professional Ltd. Reproduced with permission from Butterworth–Heinemann, Oxford. There is risk of sulfate anhydrite and deposition to the right of the vertical line spanning the whole diagram. Location in this diagram of our experimental trials and % of water content in the solution are indicated by arrows. The red arrow shows how moving in the area of interest can effect where we are in diagram.

**Figure 7 materials-14-07106-f007:**
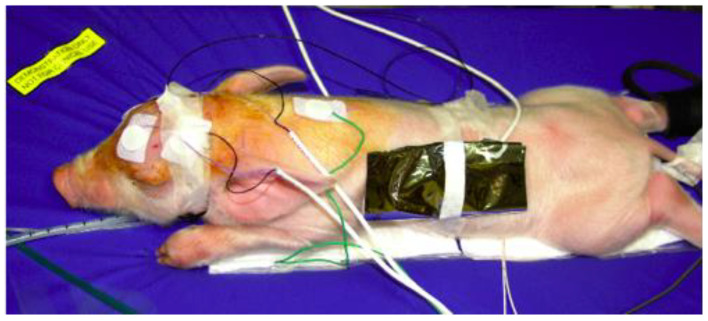
An asphyxia piglet on a PCM 32 mattress with a gel mattress cover continuously cooled to 33.5 °C. From a study Iwata et al. [45].

**Figure 8 materials-14-07106-f008:**
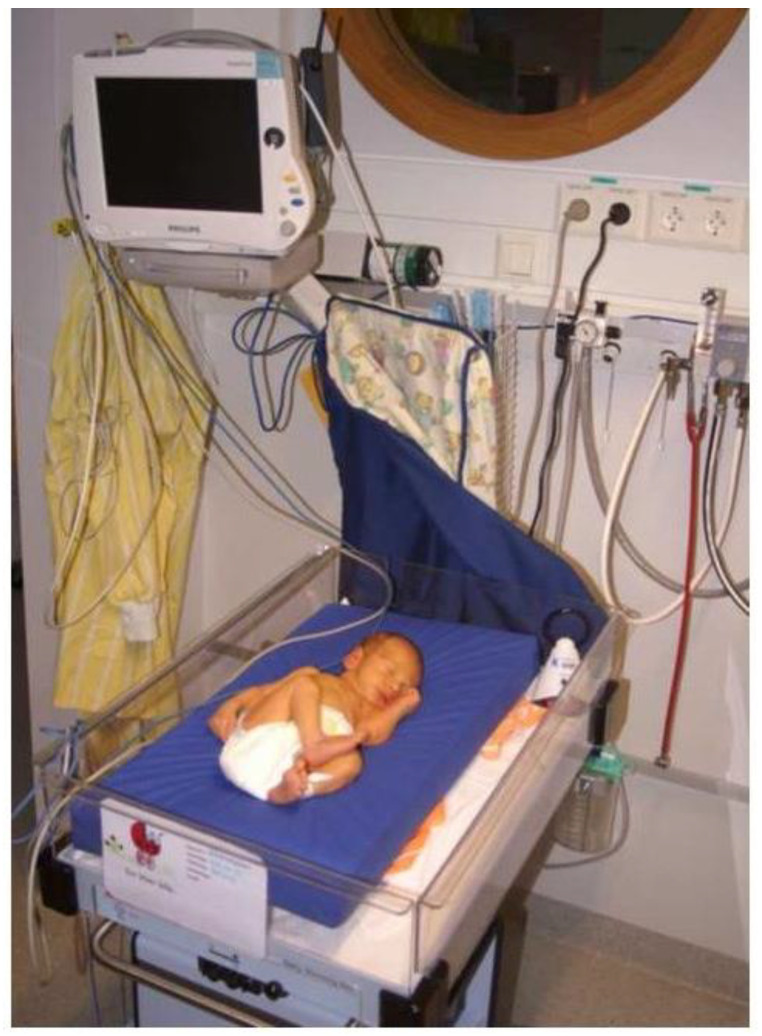
Baby being cooled on a PCM-mattress (blue) in a neonatal ward. With permission from the parents. The mattress contains PCM and a layer of soft material.

**Table 1 materials-14-07106-t001:** PCM criteria needed for medical applications such as cooling of infants and children suffering from oxygen deprivation.

No.	Criterion	What Needs to Be Done
1.	Must deliver controlled set temperature that gives the cooled object a temperature of 33.5–34.5 °C.	New mixtures that can deliver the temperatures the medical market asks for.
2.	Must be non-toxic.	The ingredients in the solution need to be tested for toxicity or the material needs to be securely encapsulated.
3.	Must not expand or in any other way cause injuries to the intended target.	Needs to be tested for each solution in a helicopter test.
4.	Must have a rapid cooling effect but with no cooling undershoot.	Needs to be tested.
5.	Should be stable for at least 72 h.	If cooling time is shorter, a fast change should be possible without problems for staff or patient.
6.	Be of low cost.	Enables use in developing countries
7.	No interbatch variability.	Each mixture needs to be tested to fulfill criteria.
8.	Able to cycle through cooling phase change at least 100 times.	Needs to be tested.
9.	Effective without electricity or water supply is a major advantage.	Choose PCM as the cooling agent.
10.	Reversed phase change no longer than 4 h.	Reversed cycle needed to be tested.
11.	Magnetic resonance imaging compatible.	No metallic or magnetic compounds can be involved in end-product.

## Data Availability

All study data can be asked for by the Corresponding Author.

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
