# Peer review of "Phase-Changing Glauber Salt Solution for Medical Applications in the 28–32 °C Interval"

_materials, 2021, doi:10.3390/ma14237106_

Round 1
Reviewer 1 Report
In the paper entailed “PHASE-CHANGEING GLAUBERSALT SOLUTION FOR MEDICAL APPLICATIONS IN THE 28-32°C INTERVAL”, the authors report the preparation process to obtain phase changing materials with medical applications in the range of 28-37.5 oC. The authors have showed that the Glauber salt solution obtained was suited for cooling of infants suffering of asphyxia.
The study gives new information on the topic of phase changing materials in general and on their clinical use in particular, however a few points need to be improved.
The authors have to carefully check the editing and phrasing. There are errors writing chemical formulas or temperature.
If the authors keep the subsection, they need to have proper titles
To make it easier for the readers, the authors should make a table reporting the conditions tested as well as some characteristics.
Considering that in discussion section are also reported results, the authors may consider to couple results and discussions in one section. Also, to make it easier, some subsections can be considered in the section.
The conclusion section is too long and parts of it can be moved to results and discussions section. A phrase indicating the best PCM solution found would be good.
The paper can be considered for publication after major revisions.
Author Response
Manuscript ID: materials-1429821 - Major Revisions
Dear Ms. Clove Liu, and reviewer 1
Thank you very much for your letter, and for allowing us to carry out major revision of our manuscript. As requested, our revisions are shown using the track changes mode. We apologize for some language errors in the previous version. The English of the entire manuscript has now been carefully revised. Below is our responses and revisions based on the reviewers comments.
Revisions and response to Reviewer 1
We thank Reviewer 1 for noting that "The study gives new information on the topic of phase changing materials in general and on their clinical use in particular" and for comments and suggestions.
- "The authors have to carefully check the editing and phrasing."
- We have carefully checked the editing and phrasing, as well as the English itself. Some sentences were too long, others not well presented. There were also some repetitive sections which have been deleted.
- "There are errors writing chemical formulas or temperature"
- We apologize and thank Reviewer 1 for pointing this out. Chemical formulas and temperature information has been corrected.
- "If the authors keep the subsection, they need to have proper titles"
- We have added titles to the subsections where needed.
- "To make it easier for the readers, the authors should make a table reporting the conditions tested as well as some characteristics."
- All key data is found in the manuscript
- "Considering that in discussion section are also reported results, the authors may consider to couple results and discussions in one section."
- We thank Reviewer 1 for this suggestion. We have now revised the manuscipt such that all results are in the Results section.
- "Also, to make it easier, some subsections can be considered in the section."
- Since we prefer to separate Results and Discussion, we think that additional subsections are not needed.
- "The conclusion section is too long and parts of it can be moved to results and discussions section."
- We thank Reviewer 1 for pointing this out. There is now better sorting out between Result and Discussion. We have also deleted a section in Discussion.
- "A phrase indicating the best PCM solution found would be good."
- We fully agree. Such a phrase has been added at the end of the Discussion section
- "The paper can be considered for publication after major revisions."
- Again, we thank Reviewer 1 for helpful comments, and hope our response and revisions are satisfactory.
Author Response
Revisions and response to Reviewer 2
We thank Reviewer 2 for valuable comments, and noting that our goal is "to obtain simple and cheap cooling apparatuses for therapeutic hypothermia" Our response and revisions are as follows:
- "the manuscript describes experiments monitoring the mixtures response to heating and cooling and presents few results of laboratory tests", and " Two Figures related to animal and human tests are also reported, but no quantitative data on the performance of the PCM-devices here used is provided"
- The purpose of the study was to find a mixture that would melt and crystallize in the clinically important 28-32°centigrades window, and that should be able to cool babies without the need of water or electricity. There was no intent to prove that this would help newborn babies at risk for hypoxy-caused brain damage. However, since time has passed, and cooling matresses with the mixture developed by us are experimentally used clinically, so we thought to include an example of the set ups for a piglet and a baby.
- "The description of the systems and methods is insufficient to reproduce these experiments and the results are reported in a quite inaccurate way."
- With due respect, all information needed about the combination of ingrediencies used and how to make the accomplished material is given.
- 3. "Moreover, additional experiments should be performed, to better evaluate the time/temperature behaviour of the systems in thermal contact with media having heat capacities similar to those encountered in medical applications."
- Each experiment was run multiple times and we only show selected curves so that the reader can get a picture. Our present study has used precision equipment to generate a cethyl methyl cellulose gelified version of Glauber salt with added salt to find a phase change material with optimal characteristics. Time aspects are shown in Fig 4.
- 4. "The equation at page 2 is not clear"
-We have updated the equation. - "The link at page 3 is not effective"
-Point taken, the link have become the ref and link have been taken away. - 6. "The exact composition of all investigated mixtures should be reported, including the one of jellified water."
-All compositions are included in the manuscript and as jellified water - "In the paragraph on the preparation (page 4 lines 149- ), also the amount of added water should be indicated."
-We have changed the text to fulfill what the reviewer thought. - "The supplier of CMC should be reported."
- The CMC was provided to us by Merck, through Royal Institute of Technology Sweden. - "What is the volume of mixture in the test tubes?"
-we used 10ml test tubes - "How were the dissolution (heating) and crystallization (cooling) temperatures detected?"
-in our model we as decribed in the manuscript we have used a computerized technology set up, with a heat bath, and heatprobes, we then have circulated each testing and then got the verified results where we can read the dissolution and crystalisation temperatures. - "Data in Figure 3 miss errors. Was the reproducibility evaluated?"
- as mentioned in the text we have run each experiment multiple times and circulated so that we could get a curve for each mixture. - "The legend of Figure 3 mentions a blue curve that is not visible."
- We are sorry for that, we have tried to send a figure where this is better visualized. - "Was there a hysteresis in the thermal behaviour of mixtures?" – after the first runs the hysteresis is clear but we only intend to use part and it is with in limits for what the intend use is.
- "Figure 4 is unclear. Variables and units of X and Y axes are missing. Moreover, the legend should report the composition of the three mixtures."
-a new and better version of the figure is uploaded. - "Abbreviations used in Figure 5 should be defined." - Thank you for the comment we have done that.
- "In page 7 line 236 the weight of water (10.33 g) is equal to the weight of Na2SO4 in page 4. Is it a mistake?" Yes.
17." In the text of page 7: Comparing our data to fig 6, our curve does not appear to fit the published diagram so well. This is due to our curve seeming to plateau, but as indicated in the curve drawn on top, the curve actually continues descending when higher amounts of NaCl is mixed into the solution. it is unclear what is the curve under discussion." – We have tried to explain this in the text. With the arrows etc.
Round 2
Reviewer 1 Report
The authors have made the changes asked and the manuscript can be published.
Author Response
Response to Reviewer 1
We thank Reviewer 1 for stating "The authors have made the changes asked and the manuscript can be published." And we thank the reviewer for his previous comments.
Reviewer 2 Report
The Authors performed a partial revision of the manuscript, which improves this contribution. I agree for publication as long as these residual points are addressed in the manuscript:
1) What I meant with previous question "How were the dissolution (heating) and crystallization (cooling) temperatures detected?" is :
To determine the value of the dissolution/ crystallization temperature, did you detect dissolution/ crystallization in your sample by a visual inspection and then you recorded the corresponding temperature? Or was the value of the dissolution/ crystallization temperature established on the basis of the behavior of the sample temperature as a function of time (there should be a change in the derivative of the curve T vs time in correspondence to a first order transition) ?
2) Concerning my previous comment: "Data in Figure 3 miss errors. Was the reproducibility evaluated?"
In Figure 3 the error bars have to be added to data, to quantify the reproducibility. Having done each experiment multiple times, you can calculate a standard deviation for the value of each melting temperature and use it as the error.
Author Response
Response to Reviewer 1
We thank Reviewer 1 for stating "The authors have made the changes asked and the manuscript can be published."
Response to Reviewer 2 We thank reviewer 2 for stating "The Authors performed a partial revision of the manuscript, which improves this contribution. I agree for publication as long as
these residual points are addressed in the manuscript:"
1a. "What I meant with previous question "How were the dissolution (heating) and crystallization (cooling) temperatures detected?" is : To determine the value of the dissolution/ crystallization temperature, did you detect dissolution/ crystallization in your sample by a visual inspection and then you recorded the corresponding temperature?
1b. "Or was the value of the dissolution/ crystallization temperature established on the basis of the behavior of the sample temperature as a function of time (there should be a change in the derivative of the curve T vs time in correspondence to a first order transition) ?"
-Thank you for your comment, we have tried to clarify this. 1a:Yes. We visual inspected the solutions as well as 1b: used a multiple run where in run included the first dissolution/ crystallization temperature and then for the diagram used later runs than 2 for points as you mentioned above for correspondence.
2. "Concerning my previous comment: "Data in Figure 3 miss errors. Was the reproducibility evaluated? In Figure 3 the error bars have to be added to data, to quantify the reproducibility. Having done each experiment multiple times, you can calculate a standard deviation for the value of each melting temperature and use it as the error."
- We deeply apologize for not having provided error bar data. As time went on, one of the collaborators died and it has been difficult to obtain these data. However, we have now calculated the correlation of the mean data points between temperature and %NaCl from 0 to 4,5 %NaCl. This correlation is strongly significant and thus supports validity of figure 2.